# How Should Patients with a Thoracolumbar Injury Classification and Severity Score of 4 Be Treated?

**DOI:** 10.3390/jcm10214944

**Published:** 2021-10-26

**Authors:** Nam-Hun Lee, Sung-Kyu Kim, Hyoung-Yeon Seo, Eric T. Park, Won-Young Jang

**Affiliations:** 1Department of Orthopaedic Surgery, Chonnam National University Medical School and Hospital, Gwangju 61469, Korea; jadorenh@naver.com (N.-H.L.); 1976skkim@daum.net (H.-Y.S.); orthopaedoc17@gmail.com (W.-Y.J.); 2Department of Orthopaedic Surgery, Emory Spine Center, Emory University, Atlanta, GA 30084, USA; 3Department of Biology, College of Arts and Sciences, Emory University, Atlanta, GA 30322, USA; eric.park@alumni.emory.edu

**Keywords:** thoracolumbar fracture, TLICS 4, treatment, nonoperative, failure

## Abstract

The thoracolumbar injury classification and severity score (TLICS) system help surgeons decide whether patients should undergo initial operative treatment or nonoperative treatment. However, the best treatment for patients with TLICS 4 fracture remains unknown. The aim of this study was to identify the risk factors for nonoperative treatment failure in patients with TLICS 4 fracture and establish treatment standards for TLICS 4 fractures. This study included 44 patients with TLICS 4 fracture who initially received nonoperative treatment. We divided these patients into two groups: the successful nonoperative treatment group included 18 patients, and the operative treatment group after nonoperative treatment failure included 26 patients. In multiple logistic regression analysis, spinal canal compromise (odd ratio = 1.316) and kyphotic angle (odd ratio = 1.416) were associated with nonoperative treatment failure in patients with TLICS 4 fracture. Other factors, including age, sex, BMI, initial VAS score, and loss of vertebral body height, were not significantly associated with nonoperative treatment failure in these patients. Spinal canal compromise and kyphotic angle were associated with nonoperative treatment failure in patients with TLICS 4 fracture. Therefore, we recommend the surgeon observe spinal canal compromise and kyphotic angle more carefully when deciding on the treatment of patients with TLICS 4 fracture.

## 1. Introduction

Thoracolumbar fracture is one of the most common injuries that cause neurological damage. These are mostly the result of trauma caused by traffic accidents or falling from heights [1,2,3]. They account for approximately 15–20% of all spine injuries [4,5]. Many classifications for thoracolumbar fractures have been studied, including Denis, AO spine, and the thoracolumbar injury classification and severity score (TLICS) classification. The main goal of these fracture classification systems is to establish a reliable universal criterion for dividing thoracolumbar fractures.

Among these classification systems, the TLICS system is frequently used by spine surgeons to categorize thoracolumbar fractures and devise appropriate therapeutic strategies. While operative treatment is typically preferred for severe fractures with posterior ligamentous complex injury or neurological deficit, nonoperative treatment is usually preferred for minor fractures with minimal bone injuries [4]. However, the TLICS system has a major drawback when it comes to deciding the treatment for patients with TLICS 4 fracture [6,7,8]. The best treatment for thoracolumbar fracture patients with moderate injury of TLICS 4 remains the surgeon’s choice [9,10]. If both operative and nonoperative treatments produce good results in patients with TLICS 4 fractures, then nonoperative treatment should be recommended over operative treatment. However, when treating patients with TLICS 4 fracture, some surgeons choose early surgery, whereas others choose nonoperative treatment initially [11]. Further study into the treatment that can provide better clinical outcomes can have a significant impact on patient care [4].

Therefore, the aim of this study was to identify the risk factors for nonoperative treatment failure in patients with TLICS 4 fracture by analyzing a retrospective case series and establish a treatment standard for TLICS 4 fracture.

## 2. Materials and Methods

We retrospectively collected the medical data of patients with acute thoracolumbar fractures who were treated at our hospital between January 1997 and December 2016. Only patients with TLICS 4 fractures who initially received nonoperative treatment were included in this study. TLICS 4 point of these patients were confirmed retrospectively during this study. At the time of treatment, the treatment method was determined according to the surgeon’s judgment by referring to the symptoms of the patient and radiological examinations.

Patients who initially received operative treatment were excluded because they could not be switched to nonoperative treatment later because of treatment failure. The exclusion criteria were as follows: patients with TLICS 4 fracture who initially received operative treatment, follow-up duration less than 2 years, serious injuries associated with other major organs, pathological fractures (e.g., tumor, infection, or osteoporosis), a history of previous spine surgery, and poor general condition. All TLICSs were re-evaluated by two spine surgeons. In case of disagreement between the two surgeons, the scores were determined in consensus via a discussion. Evaluation of interobserver reliability between the two observers was performed by using the Fleiss kappa static, ranging from −1.0 (complete disagreement) to 1.0 (complete agreement). In our study, the interobserver confidence Fleiss kappa value was 0.78.

Nonoperative treatment consisted of pain control and immobilization. The patients were initially treated with a bed rest of 3–5 days until the pain was tolerable. Compression stockings or intermittent pneumatic compression devices were used for deep vein thrombosis prophylaxis during bed rest. The patients wore the thoracolumbosacral orthosis (TLSO) for 8–12 weeks [12,13,14]. Nonoperative treatment failure was defined as the need for surgery because of persistent severe back pain [15,16,17] and/or new or worsening neurological deficits during nonoperative treatment.

The patients’ clinical data were acquired by using medical records and included their age, sex, medical history, injury mechanism, body mass index (BMI), initial visual analog scale (VAS) score, and complications. Radiological evaluations investigated spinal canal compromise (SCC), kyphotic angle (KA), and loss of vertebral body height (LOVBH). Radiological data were acquired using plain radiography and computed tomography (CT) before treatment and during follow-up. KA and LOVBH were directly measured on plain radiographs, and the percentages of LOVBH were calculated using the formulas given in Figure 1. SCC was evaluated using CT axial images, as shown in Figure 2. SCC was measured as a ratio of the canal area of the injured level to the average of that of the two adjacent intact segments.

Age, sex, BMI, initial VAS score, injury mechanism, SCC, LOVBH, and KA were selected as potential risk factors. These risk factors were compared between the successful nonoperative treatment (success group) and failed nonoperative treatment groups (failure group). This research was approved by the Institutional Review Board of the author’s university hospital, and informed written consent was waived from the participants.

SPSS 22.0 for Windows/Macintosh (IBM Corp., Armonk, NY, USA) was used for statistical analysis. A *p*-value of <0.05 was taken as statistically significant. Data were presented as mean ± standard deviation. The following tests were also conducted: χ^2^ test for categorical variables and *t*-test for continuous variables. Factors that predicted nonoperative treatment failure were analyzed using binary logistic regression.

## 3. Results

### 3.1. Demographic Data

A total of 44 patients with TLICS 4 fracture underwent initial nonoperative treatment. Of these, 18 patients only underwent nonoperative treatment (success group) (Figure 3). The remaining 26 patients (59%) received operative treatment during follow-up because of nonoperative treatment failure (failure group) (Figure 4). Among 26 patients who received operative treatment, 18 patients underwent surgery for persistent severe back pain, and 8 patients underwent surgery for new or worsening neurological deficits. Most patients wanted an operation for worsening pain and neurological changes. Patients underwent open posterior fixation and fusion or percutaneous pedicle screw fixation. There were 23 patients with TLICS 4 fracture who underwent initial operative treatment during the same period.

Demographic data of the patients are presented in Table 1. The average age was 48.6 years (standard deviation (SD) = 16.4) in the success group and 45.2 years (SD = 12.8) in the failure group. The average BMI was 23.7 (kg/m^2^) (SD = 3.2) in the success group and 22.9 (kg/m^2^) (SD = 3.5) in the failure group. No significant differences were observed between the success and failure groups in sex (*p* = 0.85), age (*p* = 0.46), BMI (*p* = 0.46), and initial VAS score (*p* = 0.11). The most common cause of fracture was falling from heights (22 patients), traffic accidents (13 patients), and blunt trauma (9 patients). No significant differences were observed between the success and failure groups in the injury mechanism (*p* = 0.92). The average time to nonoperative treatment failure was 6.8 weeks (12–67 days). The neurological changes and VAS changes of the patients are summarized in Table 2 and Table 3.

### 3.2. Radiological Outcomes

All the patients had a single-level fracture. The distribution of the 44 levels of fracture were T11 (4 patients), T12 (6 patients), L1 (20 patients), and L2 (14 patients). Radiological analysis indicated that SCC, LOVBH, and KA were significantly different between the success and failure groups. SCC and LOVBH were significantly lower in the success group (27.9 ± 7.6% and 28.6 ± 7.8%, respectively) than in the failure group (35.7 ± 13.3% and 34.1 ± 6.0%, respectively) (*p* = 0.03 and *p* = 0.01, respectively). KA was 10.1 ± 6.2° in the success group and 13.9 ± 5.0° in the failure group (*p* = 0.04) (Table 4). In 23 patients with TLICS 4 fracture who underwent initial operative treatment during the same period, the SCC was 40.1 ± 13.9%, and the KA was 16.8 ± 6.3°.

### 3.3. Statistical Analysis

Results from the simple logistic regression analysis are shown in Table 5. Significant differences were observed between the success and failure groups in the categories SCC, LOVBH, and KA (*p* < 0.05). Multiple logistic regression analysis was also performed for each category (Table 6). The result showed that SCC and KA could be considered significant risk factors for nonoperative treatment failure in patients with TLICS 4 thoracolumbar fracture (*p* < 0.05). The odd ratios of SCC and KA were 1.316 and 1.416, respectively.

Simple logistic regression analysis. Odd ratios provided with 95% confidence intervals. A *p*-value of less than 0.05 was considered significant.

Multiple logistic regression analysis. Odd ratios provide a 95% confidence intervals. A *p*-value of less than 0.05 was considered significant.

## 4. Discussion

### 4.1. Background

The TLICS system was designed to decide the classification of and treatment methods for patients with thoracolumbar fractures [17]. This system can help decision-making when choosing between nonoperative and operative treatments [18]. The three parameters of injury morphology, integrity of posterior ligamentous complex, and neurologic status are used to categorize thoracolumbar fractures. On the basis of these parameters, a score is given to each parameter, and the overall score is used to determine the appropriate therapeutic method for each patient. The treatment methods are recommended according to each score category: nonoperative treatment for scores ≤ 3, operative treatment for scores ≥ 5, and operative or nonoperative treatment for a score of 4 [19]. Patients with a score of 4 are treated at the surgeon’s choice [17].

### 4.2. Problem of TLICS Classification

The TLICS system seems a simple and reliable scoring system to help therapeutic decision-making. However, its biggest disadvantage is that the decision-making of TLICS 4 fractures, which are the most difficult to determine treatment, is at the discretion of the surgeon. Although many studies pointed to the fact that both operative and nonoperative treatments yield similar clinical outcomes in patients with TLICS 4 fracture, some studies showed that the clinical outcomes vary according to the treatment selected [20,21,22,23]. Indeed, in many patients with TLICS 4 fracture, treatment was converted to operative treatment after nonoperative treatment failure. Nataraj et al. retrospectively compared conservative and surgical treatment in patients with TLICS 4 burst fracture and reported no differences in outcomes between patients treated either conservatively or surgically [4]. In contrast, Mohamadi et al. reported better clinical and radiological outcomes in the operative group than in the nonoperative group of patients with TLICS 4 fracture [24]. Although the TLICS system provides acceptable criteria for therapeutic decision-making in patients with thoracolumbar fracture, some surgeons believe it is necessary to conduct further study to compare the outcomes of operative and nonoperative treatments in patients with TLICS 4 fracture so that the classification criteria can be improved, and the most appropriate clinical measures can be taken for patients.

### 4.3. Analysis of Our Results

Our study showed that patients with TLICS 4 fracture may require operative treatment if they have severe SCC and KA deformity after the injury. In this study, no significant differences were observed between the successful and failed nonoperative treatment groups in sex, age, BMI, and initial VAS score. However, the radiological analysis revealed that SCC, LOVBH, and KA were significantly higher in the failed nonoperative treatment group than in the successful nonoperative treatment group. Among the various radiological parameters, SCC, KA, and LOVBH were included as variables because they are the basic parameters that are generally identified in patients with spine fractures and factors that are also related to the prognosis after fracture treatment. SCC after a trauma implies the movement of posterior bone fragments into the spinal canal, and this may lead to neurological compression. Moreover, the greater the LOVBH and KA, the greater the impact on the vertebral column. This may also lead to spinal instability or intractable pain. However, in multiple logistic regression analysis, only SCC and KA were considered significant. In addition, the odd ratios of SCC and KA were 1.316 and 1.416, which is not very high. Nevertheless, we think that this study is meaningful in that variables showing statistically significant differences were presented in the absence of related studies. In the future, repeated studies with more patient data will be needed.

The choice between operative treatment and nonoperative treatment should be based on the patient’s symptoms, radiological findings, and the surgeon’s clinical judgment. Considering the lack of proper guidelines for choosing the treatment method for TLICS 4 thoracolumbar fractures [20,21,22,23,25], we tried to present such guidelines. However, we were unable to estimate the cut-off values to identify patients with TLICS 4 fracture who would have successful outcomes after nonoperative treatment. Owing to the limitation of the small sample size, we could not determine statistically significant and precise cut-off values. However, considering the mean values and SDs of the two groups, we carefully recommend operative treatment rather than nonoperative treatment for patients with TLICS 4 thoracolumbar fracture when SCC is approximately >35%, and changes in KA are approximately >14°. This would aid the surgeon during the decision-making process and prevent patient discomfort arising from changing the treatment method during the course of treatment. Of course, the cut-off values of the above risk factors have limits to their generalizability, so they must be carefully considered and judged comprehensively with the patient’s other factors. We believe that if we increase the sample size in a future study, we will be able to measure the exact cut-off values.

### 4.4. Limitation and Strength of Current Study

The limitations of this study are its retrospective design and the relatively small number of patients. Like other retrospective studies, we could not completely exclude potential confounders associated with retrospective data collection, such as the choice of the patient’s initial treatment method. However, in the absence of related studies so far, we think that this study is meaningful in that it presents variables with statistically significant differences. Therefore, the recommended criteria can be considered the basis for further study. In future studies, we will expand the number of cases to reinforce our results. We believe this study will help decide the proper treatment method for patients with TLICS 4 thoracolumbar fracture.

## 5. Conclusions

The treatment method proposed by the TLICS system for patients with TLICS 4 fracture is ambiguous and has several limitations. In our study, spinal canal compromise and kyphotic angle were associated with nonoperative treatment failure in patients with TLICS 4. Therefore, we recommend the surgeon observe spinal canal compromise and kyphotic angle more carefully when deciding on the treatment of patients with TLICS 4 fracture.

## Figures and Tables

**Figure 1 jcm-10-04944-f001:**
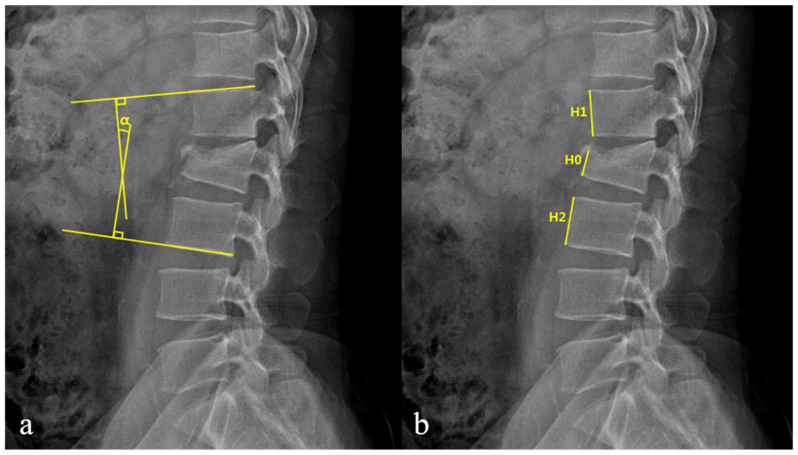
(**a**) The Kyphotic angle (KA, α) is measured by using cobb angle between the superior endplate of the upper intact vertebra and the inferior endplate of the lower intact vertebra. (**b**) The percentage of loss of vertebral body height (LOVBH) is measured by comparing the fractured anterior body height with the mean of values obtained from the upper and lower intact vertebra.; LOVBH = [(H1 + H2) − 2H0]/(H1 + H2) × 100%.

**Figure 2 jcm-10-04944-f002:**
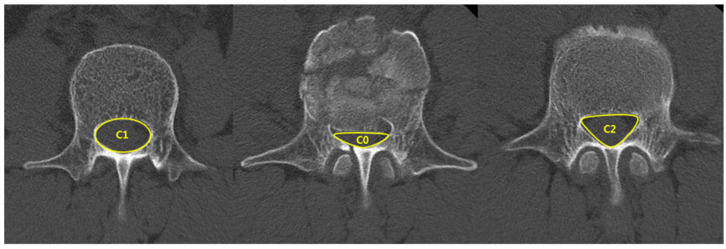
The spinal canal compromise (SCC) is calculated as a ratio of the canal area of the injured level to the average of that of the two adjacent intact segments.; SCC = ((C1 + C2) − 2C0)/(C1 + C2) × 100%.

**Figure 3 jcm-10-04944-f003:**
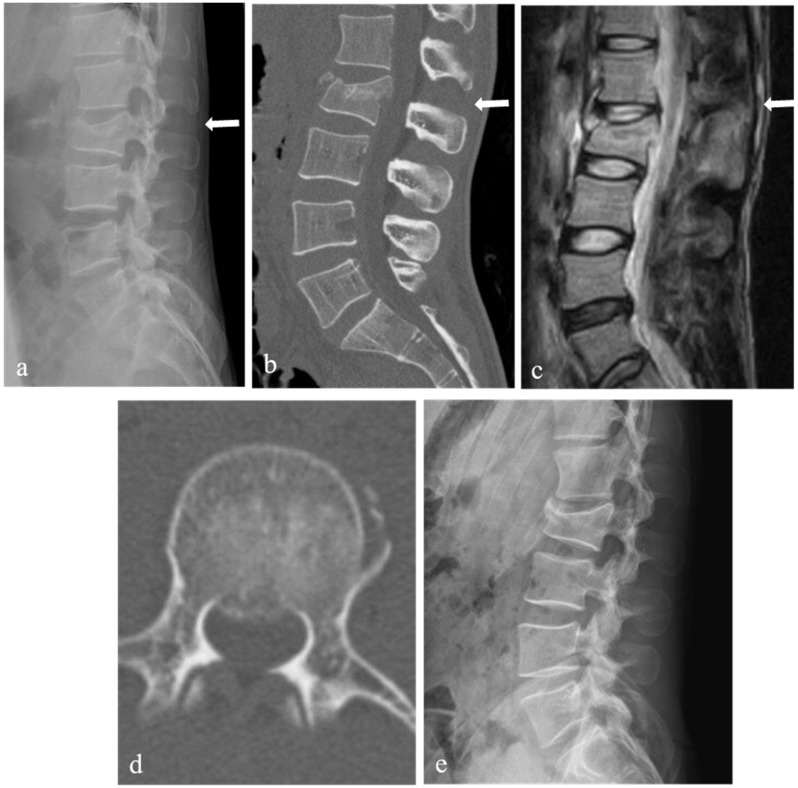
L2 burst fracture of a 39-year-old male patient with intact neurological status. Lateral X-ray (**a**) and sagittal 2-dimensional (2-D) computed tomography (CT) scan (**b**) showing L2 burst fracture with a widening of the L1-2 interspinous space (arrow), kyphotic angle of 11 degrees, and loss of vertebral body height of 31%. T2-weighted sagittal magnetic resonance imaging (**c**) showing signal change (arrow) without clear rupture of the posterior ligamentous complex. Axial CT scan (**d**) showing spinal canal compromise of 19%. At 28 months follow-up after injury, lateral X-ray (**e**) showing good spinal alignment and bony healing.

**Figure 4 jcm-10-04944-f004:**
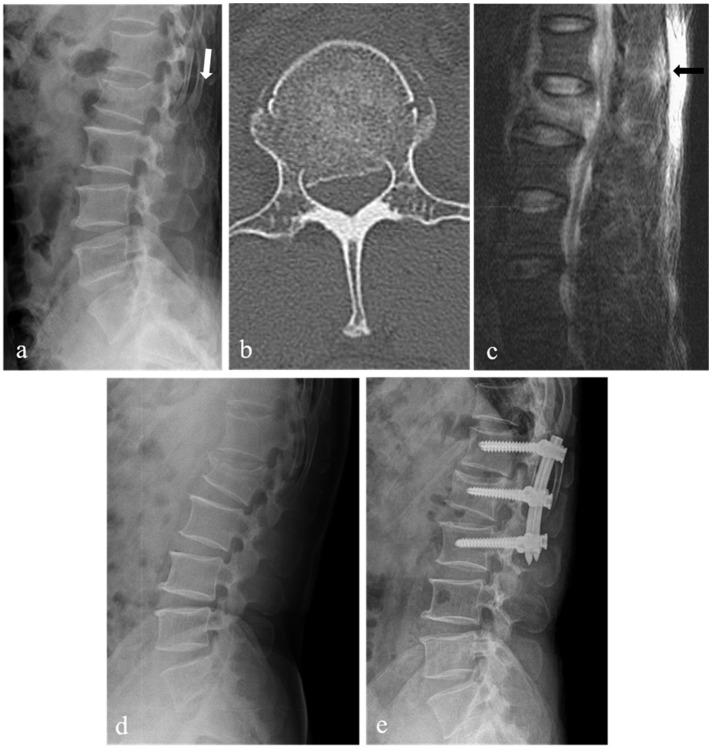
L1 burst fracture of a 52-year-old male patient with intact neurological status. Lateral X-ray (**a**) showing L1 burst fracture with a widening of the T12-L1 interspinous space (white arrow), kyphotic angle of 10 degrees, and loss of vertebral body height of 22%. Axial 2-dimensional (2-D) computed tomography (CT) scan (**b**) showing spinal canal compromise of 39%. T2-weighted sagittal magnetic resonance imaging (**c**) showing signal change (black arrow) without clear rupture of the posterior ligamentous complex. Follow-up lateral X-ray (**d**) showing the increased collapse of L1 body with a kyphotic angle of 19 degrees and loss of vertebral body height of 36%. This patient underwent surgery 12 days after injury due to rapidly progressive L1 body collapse, decreased height, and increased pain. Postoperative lateral X-ray (**e**) showing T12-L1-2 percutaneous posterior fixation.

**Table 1 jcm-10-04944-t001:** Preoperative demographic data.

	Success Group (*n* = 18)	Failure Group (*n* = 26)	*p*
Gender, male/female (*n*) ^†^	12/6	18/8	0.85
Age (yr) ^‡^	48.6 ± 16.4	45.2 ± 12.8	0.46
BMI (Kg/m^2^) ^‡^	23.7 ± 3.2	22.9 ± 3.5	0.46
Initial VAS (score) ^§^	4.5 ± 1.2	4.4 ± 0.9	0.11
Injury mechanism ^†^			
Traffic accident	6	7	0.92
Fall down	9	13	
Blunt trauma	3	6	
Level of fracture			
T11	3	1	
T12	3	3	
L1	3	17	
L2	9	5	

^†^ Pearson’s chi-square test, ^‡^ Independent *t*-test, ^§^ Man–Whitney test. Data are presented mean ± standard deviation. The *p*-value is a comparison between groups, with *p* < 0.05 indicating significance.

**Table 2 jcm-10-04944-t002:** Neurological changes of the patients.

ASIA Impairment Scale	Success Group (*n* = 18)		Failure Group (*n* = 26)		
	Initial	Last F/U	Initial	Preoperative	Last F/U
Grade A	0	0	0	0	0
Grade B	0	0	0	0	0
Grade C	0	0	0	1	0
Grade D	2	2	4	11	5
Grade E	16	16	22	14	21

**Table 3 jcm-10-04944-t003:** VAS changes of the patients.

	Success Group (*n* = 18)	Failure Group (*n* = 26)	*p*
Initial ^§^	4.5 ± 1.2	4.4 ± 0.9	0.11
Before operative treatment		6.5 ± 1.1	
Last follow-up ^§^	1.7 ± 1.1	1.9 ± 1.0	0.14

^§^ Man–Whitney test. Data are presented mean ± standard deviation. The *p*-value is a comparison between groups, with *p* < 0.05 indicating significance.

**Table 4 jcm-10-04944-t004:** Preoperative radiological data.

	Success Group (*n* = 18)	Failure Group (*n* = 26)	*p*
Spinal canal compromise (%) ^‡^	27.9 ± 7.6	35.7 ± 13.3	0.03
Loss of vertebral body height (%) ^‡^	28.6 ± 7.8	34.1 ± 6.0	0.01
Kyphotic angle (°) ^‡^	10.1 ± 6.2	13.9 ± 5.0	0.04

^‡^ Independent *t*-test. Data are presented mean ± standard deviation. The *p*-value is a comparison between groups, with *p* < 0.05 indicating significance.

**Table 5 jcm-10-04944-t005:** Simple logistic regression analysis of nonoperative treatment failure.

	Odd Ratio	95% Confidence Interval	*p*
Gender	0.9	0.25 to 3.22	0.86
Age	0.98	0.94 to 1.03	0.45
BMI	0.93	0.77 to 1.12	0.45
Initial VAS score	0.84	0.45 to 1.18	0.13
Spinal canal compromise	1.26	1.00 to 1.42	0.04
Loss of vertebral body height	1.11	1.02 to 1.15	0.02
Kyphotic angle	1.33	1.01 to 1.47	0.04

**Table 6 jcm-10-04944-t006:** Multiple logistic regression analysis of nonoperative treatment failure.

	Odd Ratio	95% Confidence Interval	*p*
Spinal canal compromise	1.316	1.01 to 1.41	0.03
Loss of vertebral body height	0.924	0.85 to 1.01	0.07
Kyphotic angle	1.416	1.02 to 1.55	0.03

## Data Availability

All data presented in this study are available on demand from the corresponding author.

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
