# Peer review of "How Should Patients with a Thoracolumbar Injury Classification and Severity Score of 4 Be Treated?"

_jcm, 2021, doi:10.3390/jcm10214944_

Round 1

Reviewer 1 Report

The authors have satisfactorily addressed my prior concerns.

Reviewer 2 Report

Lee et al. report on 44 patients with TLICS 4 fractures.

Introduction

Line 34-35: Please rewrite, "classification systems...classifying...".

Line 44-47: Please give a reference for that statement.

Why did you choose a retrospective setting instead of a prospective one?

A brief description of the current literature of secondary conversion to operative from originally nonoperative treatment is needed.

Materials and Methods

"At the time of treatment, the treatment method was determined according to surgeon’s judgement by referring to symptoms of patient and radiological examinations, not the TLICS classification" - this is problematic. Is there not a common scheme that might have been used by all surgeons? Preferably one used at your institution?

What is your reference for a thoracolumbosacral orthosis in case of a trauma?

Line 77-80 needs to be properly referenced according to national or international guidelines.

Statistics: How was tested for normality?

Results

Did you perform a power analysis before to make sure how many samples are needed for a sufficient answer to your research question?

Please focus on the reasons of secondary conversion to operative treatment.

Fig. 3C, 4C: What sequence are we looking at? Presumably T2, but please explain in the figure legend.

Tab. 5: Please indicate significant differences more clearly.

So the dependent variable for the simple logistic regression was "treatment failure"?

Discussion

The background show be a part of your introduction, as the 4.2 paragraph.

Delete "We think this topic is important because it can help surgeons make better decisions when planning treatment for patients with TLICS 4 fracture" since this is obvious.

"However, as mentioned above, this study is considered to be significant in that variables showing statistically significant differences were presented in the absence of related studies.". What does that mean? Please rewrite.

What does your study add if it was not your aim to help clinicians guide their future decisions? And why is your study not underpowered to answer that question?

Round 2

Reviewer 2 Report

The statement "However, the TLICS system has a major drawback when it domes to deciding the treatment for patients with TLICS 4 fractures" needs a reference. Please add that.

Conversion from nonoperative to operative treatment: "worsening neurological deficits" needs a definitive statment of the ASIA or VAS score.

In addition, what does "persistent severe back pain" mean, was there a certain timeline that needed to be reached before surgery was the only remaining option?

Still, please give a brief statement on each surgeons choice as whether or not to operate. A highly heterogeneous study cohort clearly limits your deduced results.

Please give latest evidence that a lumbosacral orthosis or thoracolumbosacral orthosis in case of trauma does add stability. Alternatively, add a statement limiting its abilities. I have concerns regarding that statement.

Since "Persistent severe back pain" ... can be found in guidelines as an indication for surgery, it should be easy to add as a reference.

Author Response

This manuscript is a resubmission of an earlier submission. The following is a list of the peer review reports and author responses from that submission.

Round 1

Reviewer 1 Report

Dear authors, Thank you for the contribution with your manuscript. 

The study deals with an interesting topic but the study design lacks in an important point. A retrospective study design is not recommended to answer the posed question. Rather a prospective randomized study would be necessary to answer the question.

In addition the group of thoracolumbar fractures with a TLICS Score of 4 is rather inkonsistent. Therefore ist would be recommended to include a fracture ie type A3 AO Spine in the study design.

Reviewer 2 Report

Interesting study, addresses relevant question. Spinal canal compromise, loss of vertebral body height, and kyphotic angle are markers of operative need when TLICS = 4 in a single center retrospective study of 44 patients.

Few comments for consideration:

- How did the authors select SCC >35% and KA >14% as the appropriate cutoffs for multivariate regressions (e.g. Table 4)?

- What was the dependent variable for Table 3 and Table 4 - is it "failure of conservative therapy"? If so, should include in Figure title and caption.

- Are there other clinical or radiographic parameters of interest that can be utilized to assess need for spine surgery in the trauma population? (i.e. how did the authors decide upon SCC, KA, and LOVBH as the 3 radiographic variables of interest for this study). Would be helpful to add to methods or discussion.

Reviewer 3 Report

The authors retrospectively assessed 44 patients with TLICS 4 injuries to look for factors predictive of nonoperative treatment failure. They identified that kyphotic angulation of the fracture and canal compromise were associated with nonoperative treatment failure. Overall, the manuscript addresses a pertinent topic, but I have several comments that should be addressed prior to the manuscript being suitable for publication.

  1. The manuscript requires additional grammatical editing.
  2. There may be a significant selection bias due to the retrospective nature of the study. How many patients were treated operatively at initial presentation, and how were the 44 patients selected for this study chose for nonoperative treatment initially? This limitation should be expanded upon in the discussion as well.
  3. In some cases, surgery is the “conservative” treatment option. I would recommend changing “conservative” treatment to “nonoperative” treatment throughout the manuscript.
  4. What was the average time to nonoperative treatment failure?
  5. Please report the neurologic status of the patients at initial presentation, and how that changed during follow-up leading to the need for surgery. The ASIA scale would be very helpful for presenting this data.
  6. Was there a specific kyphotic angulation was more predictive of treatment failure? The difference between the two groups in terms of kyphotic angulation, while significant, was quite small (<4 degrees), which is within the measurement error of assessment of cobb angles for kyphosis.
  7. How often were the patients assessed in follow-up and with what imaging modalities? Did the kyphotic angulation progress during follow-up?

Round 2

Reviewer 1 Report

Dear authors,

I have read your remarks but significant changes of the manuscript are not visible. The posed question/hypothesis can not be answered with the study design.